# Right Ventricular Longitudinal Strain in Patients with Heart Failure

**DOI:** 10.3390/diagnostics12020445

**Published:** 2022-02-09

**Authors:** Mengmeng Ji, Wenqian Wu, Lin He, Lang Gao, Yanting Zhang, Yixia Lin, Mingzhu Qian, Jing Wang, Li Zhang, Mingxing Xie, Yuman Li

**Affiliations:** 1Department of Ultrasound Medicine, Union Hospital, Tongji Medical College, Huazhong University of Science and Technology, Wuhan 430022, China; jimengmeng97@163.com (M.J.); wuwq1117@hust.edu.cn (W.W.); helinwh@hust.edu.cn (L.H.); glcs@hust.edu.cn (L.G.); zhangytcw@163.com (Y.Z.); linyixia@hust.edu.cn (Y.L.); qianmingzhu95@hust.edu.cn (M.Q.); 13397105628@163.com (J.W.); zli429@hust.edu.cn (L.Z.); 2Clinical Research Center for Medical Imaging in Hubei Province, Wuhan 430022, China; 3Hubei Province Key Laboratory of Molecular Imaging, Wuhan 430022, China; 4Shenzhen Huazhong University of Science and Technology Research Institute, Shenzhen 518057, China; 5Tongji Medical College and Wuhan National Laboratory for Optoelectronics, Huazhong University of Science and Technology, Wuhan 430022, China

**Keywords:** right ventricular function, longitudinal strain, heart failure, two-dimensional speckle tracking echocardiography, three-dimensional speckle tracking echocardiography

## Abstract

Patients with heart failure (HF) have high morbidity and mortality. Accurate assessment of right ventricular (RV) function has important prognostic significance in patients with HF. However, conventional echocardiographic parameters of RV function have limitations in RV assessments due to the complex geometry of right ventricle. In recent years, speckle tracking echocardiography (STE) has been developed as promising imaging technique to accurately evaluate RV function. RV longitudinal strain (RVLS) using STE, as a sensitive index for RV function evaluation, displays the powerfully prognostic value in patients with HF. Therefore, the aim of the present review was to summarize the utility of RVLS in patients with HF.

## 1. Introduction

Heart failure (HF) is a clinical syndrome with typical symptoms and signs caused by abnormal structure and function of heart [1]. HF is the end stage of various cardiovascular diseases with high prevalence, and it remains a serious public health problem [2,3,4,5]. Accurate identification of predictors of adverse outcomes of HF can help clinicians to detect high-risk patients early and provide timely intervention, which is of great clinical significance for improving long-term survival of patients. Right ventricular (RV) dysfunction is common in patients with HF [6], and it is considered to be significantly associated with the morbidity and mortality of HF [7,8,9,10]. In addition, RV dysfunction has been recognized as a powerful independent predictor of poor prognosis [11,12]. Therefore, accurate evaluation of RV function is helpful for risk stratification in patients with HF, so as to guide clinicians to choose the best treatment and improve the prognosis of patients [13,14,15,16,17,18]. Currently, cardiac magnetic resonance (CMR) imaging remains the gold standard for RV volumes and function assessment. However, the longer scan time, high cost, lack of portability and contraindications of CMR imaging hamper its routine use. Echocardiography is a first-line imaging tool for quantifying RV function in clinical practice. The conventional echocardiographic parameters of RV function recommended, such as RV fractional area change (RVFAC) [19], tricuspid annular plane systolic excursion (TAPSE) [20] and the systolic velocity of the tricuspid annulus (S’) [21] have been verified for their utilities in patients with HF. Nevertheless, there exist limitations in evaluating RV function by conventional echocardiography parameters because of the complex geometry of right ventricle [22]. In recent years, myocardial strain measured by speckle tracking echocardiography (STE) has become an established echocardiographic parameter for the diagnosis and prognostic assessments of patients with cardiovascular diseases [23]. RV strain obtained by STE, being able to detect RV subclinical dysfunction in the early stage, has been regarded as a more sensitive parameter for evaluating RV function and has important prognostic significance in patients with HF [4,16,24,25,26,27]. Therefore, RV strain suggested by the American Society of Echocardiography (ASE) has been quite useful in clinical practice as a complement to the routine echocardiography parameters, such as TAPSE or RVFAC [28]. RV strain can be classified as RV longitudinal strain (RVLS), circumferential strain and radial strain according to the direction of myocardial movement [18,29]. However, the RV wall is composed of two longitudinal myocardial fibers, the inner and the outer layers, and there is a lack of circumferential middle myocardial fibers. RVLS represents systolic shortening of the myocardial fibers and contributes to about 80% of RV function [30]. Decreased RVLS occurring in the earlier stage of RV systolic dysfunction compared with RV circumferential strain and radial strain [14,31], has significantly prognostic value in various cardiovascular diseases, thus raising awareness of the importance of RVLS assessment in patients with HF [32].

Consequently, the aim of this review was to summarize the role of RVLS assessed by 2D and 3D speckle-tracking echocardiography in patients with HF, including HF patients with preserved ejection fraction (HFpEF) and HF with reduced ejection fraction (HFrEF).

## 2. RVFAC and RVEF

Due to the complex anatomy of right ventricle, correct evaluation of its function requires using multiple parameters, such as TAPSE, S’, RVFAC, 3D RVEF, RVLS and so on. RVFAC is a significant parameter of global RV systolic function, and it is defined as: (RV end-diastolic area—RV end-systolic area)/RV end-diastolic area × 100%. RVFAC reflects longitudinal and radial information of RV systole, which overcomes the shortcoming that only confines to a single type of motion. It is associated with RVEF derived from CMR, and can provide prognostic information in patients with HF [33]. However, RVFAC still has several drawbacks which limit its clinical application. RVFAC depends on load and ignores the effect of RV outflow tract on ejection. The endocardial delineation also needs superior image quality. In addition, another main limitation of RVFAC is that the obscure definition of the RV lateral wall results to the poor interobserver reproducibility [34,35].

3D RVEF reflects longitudinal and radial components of RV systole, which overcomes the geometric limitations of traditional echocardiography parameters for evaluating RV function. 3D RVEF can explore the entire right chamber including RV outflow tract, which can include the contribution of RV outflow tract to global systolic function. It is a global measurement of RV systolic function, however, it cannot directly represent systolic performance for the reason that it depends on load and reflects the interaction between systole and load. In addition, this technique depends on adequate image quality, requires offline analysis and experience, and is time-consuming, which limit its widespread application in clinical practice [34,35].

## 3. RV Longitudinal Strain Analysis

### 3.1. Tissue Doppler Imaging

Tissue Doppler imaging (TDI) has traditionally been used to measure blood flow velocities, and it can also be used to measure myocardial and other tissue velocities, which can provide significant information about cardiac function. Strain measurement is based on myocardial velocity gradient, which is an estimate of strain rate, and therefore strain can be calculated as the temporal integral of strain rate. Strain and strain rate are measured in the apical view, where there is favorable myocardial motion along the ultrasound beam [36]. TDI-derived strain and strain rate can evaluate RV systolic function, which are not affected by geometric assumptions and have the advantages of good reproducibility and high time resolution [33,37,38,39,40]. TDI also has superiority in mature technology, including simplicity, good feasibility, and being unaffected by heart rate [41]. A variety of studies have confirmed that TDI has superiority in detecting early myocardial dysfunction [42,43]. However, TDI has major limitations that include angle dependency, low signal-to-noise ratio and the evaluation of motion information confined to myocardial segments that move along the direction of the ultrasound beam [33,37,38,39,44,45].

### 3.2. Two-Dimensional Speckle Tracking Echocardiography

In recent years, two-dimensional speckle tracking echocardiography (2D-STE) has been developed as a new imaging tool of evaluating cardiac function. 2D-STE is the two-dimensional tracking of unique speckle patterns created by the constructive and destructive interference of ultrasound beams within myocardial tissue [46]. 2D-STE allows a precise quantification of RV strain and an early detection of the subclinical RV dysfunction, providing comprehensive diagnostic and prognostic information in patients with a variety of cardiovascular diseases [16,47,48,49,50,51,52,53,54]. RVLS is measured in the RV-focused apical four-chamber view, which provides better visualization of the whole right ventricle, and avoids foreshortening of the RV apex [22]. The frame rate is between 60 and 80 frames per second. The high-quality images are essential for RV strain analysis [55]. After endocardial border delineation, the software automatically segments the right ventricle into six segments (basal, middle, and apical segments of both the RV free wall and the interventricular septum), and tracks the movement of speckles in the myocardium throughout the cardiac cycle on two-dimensional echocardiographic images [56]. Finally, RV longitudinal strain curves of free wall and septum are automatically generated by the software (Figure 1). The average value of longitudinal strain of the basal, middle, and apical segments of the RV free wall is longitudinal strain of the RV free wall (RVFWLS), whereas RV global longitudinal strain (RVGLS) represents the average value of RV six segments [57,58,59]. RVGLS may be affected by LV systolic function owing to the fact that the interventricular septum is conventionally regarded as one part of the left ventricle. Therefore, the published guideline of the ASE and the European Association of Cardiovascular Imaging (EACVI) recommend a normal value only for RVFWLS [60].

In addition to the information of myocardial motion along the direction of ultrasound beam, information on myocardial torsion, circumferential and radial motion can also be obtained by 2D-STE. STE represents a relatively rapid, high feasibility and easy-to-perform imaging tool to estimate RV function [55]. In contrast to conventional echocardiographic indices, STE also overcomes the limitations of these classical indices and provides accurate analysis of RV performance. More importantly, 2D-STE has the advantages of angle-independence and being less affected by reverberations, sidelobes and drop out artifacts compared with tissue Doppler-determined parameters. Moreover, STE-derived RV strain displays good reproducibility and less load-dependence [40,45,53,57,58,61,62,63]. Therefore, STE represents a valuable noninvasive technique that may potentially broaden our knowledge on RV function.

However, intrinsic limitations of 2D-STE should be addressed. First, 2D-STE analysis requires good image quality, which produces less accuracy in endocardial border definition and tracing in patients with suboptimal image quality. Second, cardiac motion is three-dimensional in nature, and the myocardial deformation in 2D plane measured by 2D-STE has out-of-plane motion of speckles. Moreover, the software for RV strain analysis that we use are designed for left ventricle (LV) strain evaluation; the dedicated software for RV strain measurements has not yet been released. Finally, RV strain values depend on vendor and software versions [64].

### 3.3. Three-Dimensional Speckle Tracking Echocardiography

More recently, three-dimensional speckle tracking echocardiography (3D-STE) has been introduced as a novel technique than can track the myocardial motion within the 3D volume. Thus, 3D-STE is free of geometric assumptions and out-of-plane motion of the speckles, allowing a more accurate and comprehensive evaluation of myocardial function owing to overcoming the limitations of 2D-STE [35,38,65,66]. Three-dimensional STE analysis for the right ventricle is shown in Figure 2. Its accuracy and reproducibility in assessing RV function have been confirmed in patients with transplanted hearts, pulmonary hypertension, and hypoplastic left heart syndrome after Fontan palliation [63,67,68,69]. Moreover, we previously investigated the feasibility and accuracy of 3D-STE for the quantification of RVLS in comparison with CMR imaging in a large number of study populations with a wide variety of RVEF and cardiovascular pathologies, and found that the 3D-RVFWLS values correlated better than 2D-RVFWLS values with CMR values (0.85 vs. 0.64) with smaller bias and narrower limits of agreement. Our findings demonstrated the superiority of 3D-RVFWLS over 2D-RVFWLS in evaluating RV function against CMR imaging [70,71]. However, 3D-STE also has several limitations, such as low temporal resolution and dependence on image quality [35,64].

### 3.4. Cardiac Magnetic Resonance

The main methods of evaluating RV strain by CMR imaging include CMR tissue tagging, CMR feature tracking (FT) and CMR tissue tracking (TT). FT is an image postprocessing technology which is applied retrospectively to routinely acquired cine images for strain measurements. After user-defined epicardial and endocardial borders are delineated, endocardial features were tracked by FT software. Except for GLS, which is measured on three long-axis cine images, both global circumferential and radial strain are derived from short-axis cine images. The structure and function of myocardium can be evaluated from different directions by CMR, which has the advantages of multi-parameters and high resolution and is considered as the gold standard for the assessment of the structure and function of right ventricle [14]. Yet, CMR has significant limitations, such as long scanning time, high price and contraindication with metal implant or claustrophobia. In addition, pixel size (displacement smaller than the pixel size may not be detected), artifacts from through-plane motion, and 2D tracking also limit its wide clinical application [72,73].

## 4. RV Mechanisms in Patients with HF

Impaired LV diastolic function is the main manifestation in the early stage of HF, while the systolic function is normal or slightly decreased, namely HFpEF (LVEF ≥ 50%); with progress of the disease, cardiac function is gradually decompensated and LV systolic function is impaired, namely HFrEF (LVEF < 40%) [74,75]. When LV dysfunction occurs in patients with HF, RV function is impaired as well [76]. Currently, the mechanisms of RV dysfunction in patients with HF remains extremely complex, and they can be summarized as follows: ① Pulmonary arterial hypertension. The main cause of RV dysfunction in patients with HF is chronic pulmonary venous hypertension caused by LV systolic dysfunction, which leads to the increase in RV afterload and hence the impaired RV function [25]; ② Interventricular interaction. The interdependence between left and right heart during systole and diastole has been considered as one of the main mechanisms of RV dysfunction. Because the left and right ventricle share the interventricular septum and LV contraction can provide 20% to 40% of the prime force for RV contraction, RV function can be influenced by the left ventricle through interventricular interaction when LV dysfunction occurs [14]; ③ Myocardial injury. Patients with HF are often accompanied by coronary heart disease, hypertension, obesity, diabetes, chronic obstructive pulmonary disease and other complications. The myocardium can be directly or indirectly damaged through the systemic pathway by these diseases, which usually lead to myocardial hypertrophy and myocardial fibrosis (MF), resulting in RV myocardial dysfunction [77]; In addition, neurohumoral activation in left-sided HF may have adverse influences on RV function. ④ Atrial fibrillation. Shortened ventricular filling time resulting from atrial fibrillation can give rise to the increased left atrial pressure load, which cause an increase in pulmonary vein and pulmonary capillary pressure, followed by an increase in pulmonary artery pressure and RV afterload [77].

## 5. RV Longitudinal Strain in HFpEF

RV dysfunction is common and often leads to poor prognosis in patients with HFpEF. Based on the different research methods, sample sizes and diagnostic criteria, the prevalence of RV dysfunction in patients with HFpEF fluctuates greatly. However, at least 20% of patients with HFpEF are known to have RV dysfunction, and the incidence of RV dysfunction can be potentially up to 30% to 50% [73,75,76]. The current results regarding RV longitudinal strain in patients with HFpEF are summarized in Table 1. A study [78] that involved 86 patients with HFpEF compared RVLS assessed by 2D-STE with conventional echocardiographic parameters such as TAPSE and S’ in patients with HFpEF. Additionally, the left ventricular diastolic dysfunction (LVDD) of patients with HFpEF were graded as normal LV diastolic function group, LVDD 1 grade group, LVDD 2 grade group and LVDD 3 grade group, according to the guideline of ASE in 2016. They found that 2D-RVFWLS decreased progressively among LVDD 1 grade group, LVDD 2 grade group and LVDD 3 grade group. 2D-RVFWLS was moderately related to diastolic function, and weakly correlated with age, B-type natriuretic peptide, RVFAC and TAPSE. This study demonstrated that 2D-RVFWLS was a sensitive parameter for detecting subtle changes in RV systolic function at the early stage when mild-to-moderate LV diastolic dysfunction occurred. In a study of 201 patients with HFpEF, Morris et al. [79] showed that 2D-RVGLS was lower in patients with HFpEF than in asymptomatic patients with LV diastolic dysfunction. The incidence of RV longitudinal dysfunction was 75% in patients with HFpEF. Furthermore, reduced 2D-RVGLS was significantly associated with worse New York Heart Association functional class in patients with HFpEF.

Lejeune et al. [80] found that 28 (19%) patients with HFpEF had impaired 2D-RVGLS (>−17.5%), and 2D-RVGLS was significantly altered in patients with HFpEF compared with control subjects. They confirmed that 2D-RVGLS was a powerful independent predictor of overall mortality and HF rehospitalization in patients with HFpEF. Their findings showed the superiority of 2D-RVGLS over RVFAC, TAPSE and other conventional echocardiographic parameters in predicting clinical end-point events in patients with HFpEF. In addition, 2D-RVGLS was found to be better correlated with CMR-derived RVEF than other conventional echocardiographic parameters, and therefore could more accurately quantify RV function.

Given the complex geometry of the right ventricle, RV longitudinal strain may be most accurately evaluated by 3D-STE. Until now, there have been limited studies regarding the utility of 3D-STE in RV assessment in clinical practice. A study [81] by Meng et al. comprehensively assessed prognostic value of RV function by 2D- and 3D-STE, and conventional echocardiography in patients with HFpEF. HFpEF patients with end points had lower RVEF and 3D-RVFWLS, nevertheless, 2D-RVFWLS was not different compared with those without end points. They also found that both 2D-RVFWLS and 3D-RVFWLS were able to dependently predict adverse clinical events in patients with HFpEF. Furthermore, 3D-RVFWLS provided a comparable prognostic value to 2D-RVFWLS in patients with HFpEF.

Kucukseymen et al. [82] evaluated RVFWLS in 203 patients with HFpEF by CMR-FT, and they showed that an explainable machine learning model using noncontrast CMR parameters could identify HFpEF patients at high risk of HF-related hospitalization. Additionally, RV and LA strain parameters could provide the greatest value for predicting the adverse end-point event, HF-related hospitalization, among noncontrast CMR parameters. Kammerlander et al. [83] demonstrated that the event incidence of HFpEF patients with a RVGLS that were above the median were higher. Moreover, RVGLS was still independently associated with adverse outcomes after correcting for risk factors (age, diabetes, renal function, N-terminal pro–b-type natriuretic peptide serum concentration, and RV size and function) by multivariable Cox regression analysis. Therefore, RVGLS derived from CMR-FT could provide significant value for predicting cardiovascular events in patients with HFpEF.

## 6. RV Longitudinal Strain in HFrEF

RV dysfunction is often present in patients with HFrEF. The current findings regarding RV longitudinal strain in patients with HFrEF are shown in Table 2. Houard et al. [13] compared the prognostic value of 2D-RVGLS and 2D-RVFWLS by STE against CMR-RVEF and CMR-FT-GLS and other conventional echocardiographic parameters in 266 patients with HFrEF. They revealed that 2D-RVGLS and 2D-RVFWLS could predict overall and cardiovascular mortality in patients with HFrEF. In addition, CMR-FT-RVGLS was also found to have obvious additional prognostic value for overall and CV mortality. Additionally, this study revealed that 2D-RVGLS and 2D-RVFWLS provided incremental predictive value over CMR-RVEF, CMR-FT-RVGLS, TAPSE, and FAC, suggesting that the 2D-STE parameters played an important role in risk stratification of patients with HFrEF. A study that included 332 outpatients with HFrEF in a stable clinical condition showed that both 2D-RVGLS and 2D-RVFWLS were significant predictors of all-cause mortality in univariate as well as multivariate analysis. Anther observation revealed that 2D-RVGLS and 2D-RVFWLS were independently associated with the risk of death, cardiovascular death and heart transplantation and/or death due to HF worsening in clinically stable HF outpatients [14]. Another study that included 171 patients with chronic systolic heart failure (LVEF ≤ 35%) showed that impaired 2D-RVGLS was related to increasing New York Heart Association class and greater LV volume. Worse 2D-RVGLS was also associated with decreased LVEF, impaired LV diastolic dysfunction (E/e’ septal and left atrial volume index), and conventional parameters of RV systolic and diastolic dysfunction (RV S’, e’/a’, right atrial volume index) [84].

RV strain was a significant predictor of long-term adverse outcomes. Moreover, RV strain ≥ −14.8% independently predicted adverse events after adjustment for age, LVEF, RV s’, E/e’ septal, and right atrial volume index [84]. Therefore, worse 2D-RVGLS provided incremental prognostic value to LV function in patients with HFrEF. In a study of 288 outpatients with stable HFrEF, Carluccio et al. compared the prognostic power of 2D-RVFWLS with 2D-RVGLS, and showed that both 2D-RVGLS and 2D-RVFWLS were associated with outcomes by univariable analysis [85]. Additionally, after correction for New York Heart Association class, EMPHASIS risk score, natriuretic peptides, and therapy, the relationships remained remarkable. However, in the multivariable Cox regression, the prognostic value of 2D-RVGLS was not dependent of LV systolic dysfunction [85]. 2D-RVFWLS remained independently related to a higher risk of adverse clinical events after controlling for clinical and echocardiographic predictors, including LV strain [85]. Therefore, 2D-RVFWLS offered superior predictive value over 2D-RVGLS in patients with HFrEF [85]. In a small cohort of 47 patients with HFrEF referred for cardiac transplant assessment due to refractory heart failure, Cameli et al. simultaneously performed right-heart catheterization and echocardiography examinations, and found that TAPSE or tricuspid s’ was not associated with RV stroke volume [86]. However, 2D-RVFWLS had a close negative correlation with RV stroke work index. In addition, 2D-RVFWLS showed the highest diagnostic performance with good sensitivity and specificity to predict depressed RV stroke work index [86]. A study [87] that included 20 HF patients with LV systolic dysfunction compared the test–retest reproducibility among the parameters of RV function, and it showed that 2D-STE-RVGLS was moderately associated with CMR-FT-RVGLS, however, as for the assessment of RV systolic function, 2D-STE-RVGLS demonstrated better test–retest reliability compared wih CMR-FT-RVGLS. In addition, this study found that 2D-STE-RVGLS could more accurately evaluate the strain of right ventricle and assess RV function.

## 7. RV Longitudinal Strain in HF Patients with Preserved Traditional RV Function Parameters

Several studies have reported the predictive value of RV longitudinal strain in HF patients with preserved conventional RV systolic function parameters, such as TAPSE and S′. However, these traditional RV function indicators could not accurately assess and represent global RV systolic performance because they only reflected segmental function of the right ventricle [35,88]. RVLS, as a new echocardiographic parameter, overcomes the limitation of angle dependency of TAPSE and S′, and it has incremental clinical value in detecting subclinical RV dysfunction. The current findings regarding RV longitudinal strain in patients with preserved traditional RV function parameters are depicted in Table 3. Morris et al. [89] compared 218 patients with HFpEF and 208 patients with HFrEF with controls and measured RVGLS and RVFWLS by 2D-STE. They found that the normal range of RV systolic strain in the normal population was as follows: 2D-RVGLS −24.5 ± 3.8% and 2D-RVFWLS −28.5 ± 4.8%. They demonstrated that the conventional echocardiographic parameters such as TAPSE, S’, RVFAC in patients with HFpEF or HFrEF were within the normal range, but both 2D-RVGLS and 2D-RVFWLS decreased. Their findings indicated that 2D-RVGLS and 2D-RVFWLS were related to the clinical status of patients with HF. More importantly, subtle RV longitudinal systolic dysfunction can be detected by both 2D-RVGLS and 2D-RVFWLS in a considerable proportion of patients with HFrEF and a smaller group of patients with HFpEF despite preserved TAPSE, RVFAC and S’. Carluccio et al. [24] demonstrated that 2D-RVFWLS is impaired in patients with HFrEF with preserved TAPSE. According to the lasso-penalized Cox-hazard model, impaired 2D-RVFWLS was an independent predictor of death and HF rehospitalization, providing incremental prognostic value over TAPSE and other recognized clinical and echocardiographic predictors of end points and improving risk stratification. The best cutoff value of 2D-RVFWS for prediction of outcome was −15.3%. The results of this study also demonstrated the potential of RVLS in detecting subtle RV systolic dysfunction. Three case examples of altered RV strain despite preserved TAPSE and S-TDI in patients with HFpEF, HFmrEF and HFrEF are shown in Figure 3.

## 8. RV Longitudinal Strain in Acute HF

Repeated rehospitalization and adverse outcome are usually experienced by patients with HF, and therefore, it is vital to identify vulnerable HF patients at high risk of poor prognosis among patients hospitalized because of acute HF. The current results of RV longitudinal strain in acute HF are depicted in Table 4. A study [90] that involved 618 patients hospitalized for acute decompensated HF found that only impaired 2D-RVFWLS was an independent predictor of poor prognosis in patients with acute HF in multivariate Cox models, providing incremental prognostic value over clinical and conventional echocardiographic parameters. The prognostic implication could be evidently improved by the addition of 2D-RVFWLS to clinical risk evaluation (age, New York Heart Association class III/IV, brain natriuretic peptide, and blood urea nitrogen), and the net reclassification improvement could be increased by 0.30 (*p* = 0.01). Meanwhile, left-sided indices cannot provide prognostic information of unfavorable outcomes. Hence, 2D-RVFWLS played a crucial role in identifying patients with acute HF at higher risk for cardiac events after discharge. Additionally, the author also demonstrated that 2D-RVFWLS was a more accurate marker of RV systolic function than 2D-RVGLS. Yao et al. [91] found that acute myocardial infarction group had lower 2D-RVGLS, 2D longitudinal strain of the basal and mid segments of the free wall and septum, and 2D longitudinal strain rate of the basal and mid segments of the free wall than control group. For 2D-RVGLS, the 2D longitudinal strain of the mid and apical segments of the free wall and the 2D longitudinal strain rate of the mid segments of the septum were lower in RV myocardial infarction group compared with LV inferior myocardial infarction group. Additionally, in the single-parameter mode of ROC curve analysis, the diagnosis efficiency of 2D-RVFWLS and 2D-RVGLS, whose cut-off value were −18.26% and −16.27%, respectively (sensitivity: 100% and 100%, respectively; specificity: 65.7% and 72.4%, respectively), were higher than those of other clinical and echocardiographic parameters. This study showed that RV dysfunction in patients with acute LV inferior wall myocardial infarction or complicated with RV myocardial infarction could be assessed early and accurately by 2D-RVLS, which improved the diagnosis efficiency and offered evidence for clinical evaluation of RV function. Borovac et al. [92] showed that 2D-RVFWLS had a good correlation with conventional RV functional indices (TAPSE, RVFAC and S’) in patients with acute worsening of HF. In a cohort of 1824 patients with acute HF, Park et al. revealed that 2D-RVGLS were significantly associated with all-cause mortality [93]. Patients with decreased biventricular longitudinal strain showed the worst prognosis [93].

## 9. RV Longitudinal Strain in Nonischemic Dilated Cardiomyopathy

Nonischemic dilated cardiomyopathy (NIDCM) is the most common form of cardiomyopathy, and the major cause of HF in addition to hypertension and coronary diseases. It is defined as LV systolic and diastole dysfunction in the absence of load abnormality or coronary artery disease [94]. RV dysfunction is a significant predictor of adverse outcomes in these patients [95,96]. The current findings of RV longitudinal strain in non-ischemic dilated cardiomyopathy are depicted in Table 5. Vîjîiac et al. [97] assessed the extent of RV dysfunction and assessed the ability of 2D-RVLS to predict the prognosis in 50 patients with NIDCM. 2D-RVGLS, 2D-RVFWLS and 3D-RVEF of patients with adverse cardiovascular events were obviously more impaired. 2D-RVGLS and 3D-RVEF were independently associated with outcomes after adjustment for age and New York Heart Association classification by Cox proportional hazards multivariable analysis. Although 2D-RVGLS was not independently related with adverse events after further adjustment for LV diastolic dysfunction, which is perhaps because the left atrium is the pathophysiological link between LV diastolic dysfunction and its effect on the pulmonary vascular bed and 2D-RVGLS still could provide certain value for detecting more subtle RV dysfunction than 3D-RVEF. A previous study [98] demonstrated that RVFAC, TAPSE, and 2D-RVLS were independently associated with prognosis even after adjustment for clinical and echocardiographic parameters in NIDCM patients. In addition, the joint assessment using RVFAC and RVLS could provide better prognostic value than other single or combined parameters for stratifying high-risk patients with NIDCM. Liu et al. [99] assessed the peak of RVGLS(RVpGLS) in 192 patients with C or D HF with NIDCM without atrial fibrillation by CMR-FT, and they found that RVpGLS was significantly associated with adverse outcomes in multivariate Cox regression model after adjusting for traditional risk variables. In addition, they demonstrated that patients with RVpGLS ≥ −8.5% showed poorer clinical outcomes than those with RVpGLS < −8.5% (*p* = 0.0037).

## 10. RV Longitudinal Strain and Myocardial Fibrosis in Patients with End-Stage HF

MF is a common pathological manifestation in patients with end-stage HF, and is able to result in cardiac remodeling, increased myocardial stiffness, and pump failure. Therefore, early and accurate identification of MF has significant clinical implications for assessing the progress of the disease, response to the treatment, and prognosis in patients with HF. The current results of RV longitudinal strain and myocardial fibrosis in patients with end-stage HF are shown in Table 6. A study [100] investigating 20 end-stage dilated cardiomyopathy patients showed that 2D-RVFWLS was not correlated with the MF. Lisi et al. [101] found that in 27 patients with end-stage HF, RV MF was strongly correlated with 2D-RVFWLS (r = 0.80), weakly with TAPSE (r = −0.34), sphericity index (r = 0.42) and right atrial longitudinal strain (r = −0.37). 2D-RVFWLS was not only a major echocardiographic parameter that independently correlated with MF and functional capacity, but also the most accurate echocardiographic index for detecting severe MF. The above inconsistent results regarding the correlation between 2D-RVFWLS and MF are needed to be further verified in studies on larger sample sizes. Tian et al. [102] investigated 102 patients with end-stage HF undergoing heart transplantation by 3D-STE, and showed that the severe MF group displayed lower RV stroke volume, 3D-RVEF, 2D-RVFWLS, and 3D-RVFWLS than the mild and moderate MF groups. Significantly, 3D-RVFWLS was correlated better with the degree of RV MF than 2D-RVFWLS and other conventional RV function parameters. Furthermore, 3D-RVFWLS was better in predicting the extent of RV MF than 2D-RVFWLS in multivariate stepwise linear regression analysis. This study indicated that 3D-RVFWLS was the most reliable echocardiographic index to predict the degree of RV MF in patients with end-stage HF. Therefore, 3D-RVFWLS could be considered as a novel imaging marker for evaluating RV MF and was useful in guiding optimal therapy in patients with HF.

## 11. RV Failure Following LV Assist Device Implantation

LV assist device (LVAD) is one of the treatments for patients with end-stage HF. As a bridge to transplantation, LVAD helps to improve organ function, quality of life and clinical outcomes. RV failure following LVAD implantation remains a major cause of higher morbidity and mortality [16,103,104,105]. The current findings of RV longitudinal strain in RV failure following LV assist device implantation are shown in Table 7. Dufendach et al. [15] demonstrated that reduced 2D-RVFWLS before LVAD implantation was associated with higher incidence of RV failure after the implantation of a LVAD. Therefore, 2D-RVFWLS was a powerful predictor of RV failure following LVAD implantation. Using the optimal cutoff value of −5.64%, the c-index of 2D-RVFWLS in predicting RV failure was 0.65. In contrast, 2D-RVGLS was not predictive of RV HF after LVAD implantation, as 2D-RVGLS was mainly affected by the interventricular septum. Wei et al. [106] also showed that 2D-RVFWLS was an independent predictor of RV dysfunction in patients receiving LVAD implantation, suggesting it plays an important role in predicting patients who may develop RV failure. Another investigation revealed that 2D-RVFWLS was associated more strongly with RV failure following LVAD implantation than TAPSE or Michigan risk score, indicating the independent and incremental role of 2D-RVFWLS for the prediction of RV failure development after LVAD implantation [107]. In a study of a small number of patients receiving LVAD implant, Cameli et al. [108] investigated the predictive value of RVLS in these patients. They revealed that patients with lower 2D-RVFWLS at preoperative assessment had a worse prognosis. In contrast, patients with higher 2D-RVFWLS at baseline displayed an improvement in RV deformation. In a meta-analysis of 4428 patients who underwent LVAD implantation, 2D-RVFWLS had the highest effect size in predicting RV failure after implantation. Patients with lower RV strain values can identify patients at higher risk for post-LVAD RV failure [109]. Until now, 3D-STE data in HF patients receiving LVAD implantation are scarce. A recent study examined patients who had RVFWLS compared with those without RV failure [104]. ROC analysis revealed that 3D-RVEF and 3D-RVFWLS displayed high discriminative capabilities in detecting RV failure. More importantly, 3D-RVEF and 3D-RVFWLS are independently associated with RV failure and adverse clinical outcomes in patients undergoing LVAD implantation [104]. Thus, RV longitudinal strain is extremely crucial for the risk stratification of patients who undergo LVAD implantation.

## 12. RV Contractile Reserve in HF

RV contractile reserve is a new echocardiographic parameter that has emerged in recent years, which is also of vital importance to evaluating the outcomes of HF patients. It is defined as the capability of the RV to adapt to the increased pulmonary vascular resistance and is regarded as a sensitive index of volume load, which reflects the dynamic function of the right ventricle [110,111]. It is expressed by a variety of parameters, which may be determined by right-heart catheterization as an invasive method or noninvasively by stress-Doppler-echocardiography [112]. A study [113] that involved 16 patients with severe HF researched the RV contractile reserve of these patients by assessing the response to inotropic modulation with the $\beta_{1}$-agonist dobutamine. They found the capability of increasing RV performance in response to increased dobutamine infusion was related to the short-term clinical outcomes, and patients who were incapable of increasing ventricular function in response to an increase in dobutamine infusion displayed risk of further deterioration of their condition. This study demonstrated that RV contractile reserve could provide prognostic value in patients with severe HF. Kinoshita et al. [114] examined RV contractile reserve in 67 patients with HF by low-load exercise stress echocardiography (ESE). The increases in RV systolic(s′) velocity, TAPSE, and RV strain during low-load exercise were defined as the indicators of RV contractile reserve. The variation in RV s′ velocity was significantly associated with peak oxygen uptake (${VO}_{2}$), a crucial prognostic parameter in patients with HF, during low-load ESE. The variation in RV s′ velocity during low-load ESE was an important indicator for assessing exercise capacity in HF patients. They found that the evaluation of RV contractile reserve was associated with the clinical outcomes of patients with HF, and it could contribute to distinguishing HF patients with high risks.

## 13. RV Longitudinal Strain in Congenital Heart Diseases

In recent years, RV strain has been regarded as a new echocardiographic parameter for the prognostic assessments of patients with cardiovascular diseases [23], which include congenital heart diseases. The current results of RV longitudinal strain in congenital heart diseases are summarized in Table 8. Almeida-Morais et al. [115] evaluated the relationship between 2D-RVLS and RV function parameters in 42 patients with repaired tetralogy of Fallot. 2D-RVLS was correlated with TAPSE, RVEF, RVFAC, pulmonary regurgitation color length, RV end-systolic volume and LVEF. Additionally, 2D-RVLS and LVEF were independently related to RVEF. They also demonstrated that 2D-RVLS was significantly associated with RVEF measured by MRI, and in the multivariate analysis, it was the only echocardiographic parameter that related to RVEF. 2D-RVLS could contribute to assessing RV function, and could help to predict long-term outcomes in patients with tetralogy of Fallot. In a retrospective analysis [116] that involved 26 patients with transposition of the great arteries, investigators evaluated RV function using 2D-STE in these patients. The study demonstrated that the evaluation of RV function by 2D-RVLS was a simple and feasible method in patients with transposition of the great arteries, and it could provide prognostic significance in serial follow-up. Steinmetz et al. [117] evaluated RV myocardial deformation in 30 patients with Ebstein’s Anomaly (EA) and 20 healthy control subjects by CMR-FT. The RVGLS was impaired in patients with EA compared with healthy subjects, and it was significantly correlated with HF parameters, right/left-volume Index, New York Heart Association classification and brain natriuretic peptide. They demonstrated that involving RVGLS in CMR evaluation may contribute to detecting cardiac deterioration in the early stage and improving clinical management and decision making in EA patients.

## 14. RV Longitudinal Strain in Chemotherapy Cardiotoxicity

Chemotherapy is one of the most effective treatments of cancer at present. However, cardiotoxicity is one of the most devastating complications of cancer treatment and has become the main risk factor for prognosis of cancer patients. Therefore, careful monitoring of ventricular function plays important role in the treatment process [118]. The current results of RV longitudinal strain in chemotherapy cardiotoxicity are summarized in Table 9. Wang et al. [119] studied RV function after anthracycline exposure in 61 patients with diffuse large B-cell lymphoma (DLBCL) treated with anthracycline. The proportion of RV systolic dysfunction was obviously higher in patients with cardiotoxicity compared with patients without cardiotoxicity. In addition, 2D-RVFWLS and 3D-RVEF were superior parameters in detecting subtle RV dysfunction and predicting outcomes over 2D-RVGLS and conventional echocardiographic parameters. Decreased percentages of 2D-RVFWLS and 3D-RVEF were independently associated with anthracycline-related RV systolic dysfunction in DLBCL patients receiving chemotherapy. They confirmed that the assessment of RV function should be involved in the routine clinical follow-up during the cancer treatment process to improve the evaluation of prognosis and therapeutic management of DLBCL patients. Keramida et al. [120] studied 101 HER2-positive breast cancer women on trastuzumab treatment, and they found that trastuzumab may result in subclinical myocardial damage on the right ventricle. Additionally, a percent change of −14.8% in 2D-RVGLS could predict cardiotoxicity with 66.7% sensitivity and 70.8% specificity (area under the curve 0.68, 95% confidence interval 0.54–0.81), which was able to correctly classify 90% of women with cardiotoxicity. RVLS had superiority in detecting chemotherapy cardiotoxicity at an earlier stage than conventional echocardiographic parameters.

## 15. Conclusions

RV function can offer important clinical and prognosis information in various clinical settings. RVLS, as a new echocardiographic parameter, provides a more comprehensive and detailed evaluation of RV systolic function, allowing for an early detection of subclinical RV dysfunction. Furthermore, RVLS has been regarded as an independent predictor of poor prognosis in patients with HF, regardless of LVEF. RVLS also can provide superior value of prognosis in nonischemic dilated cardiomyopathy, congenital heart diseases and chemotherapy cardiotoxicity. Thus, it is a powerful index for identifying patients who are at higher risk among HF patients and other cardiovascular diseases. Nevertheless, further studies are essential to determine its role in the management of patients.

## Figures and Tables

**Figure 1 diagnostics-12-00445-f001:**
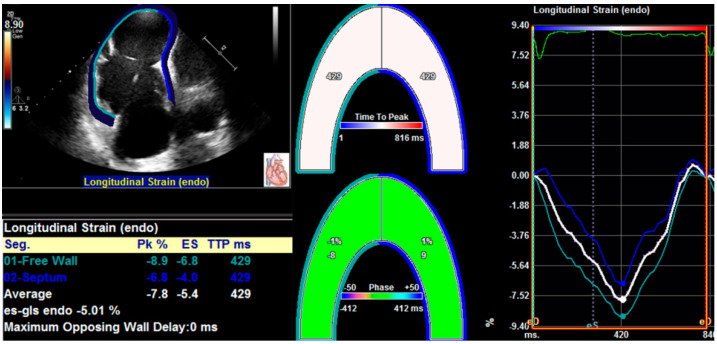
Longitudinal strain of the RV free wall and septum using two-dimensional speckle tracking echocardiography.

**Figure 2 diagnostics-12-00445-f002:**
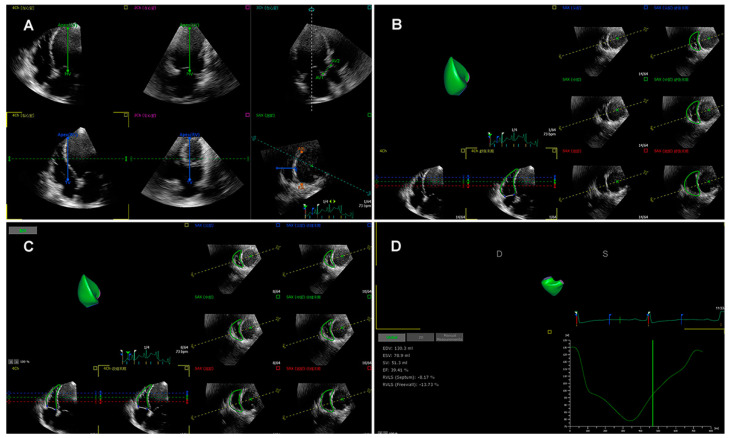
Longitudinal strain of the RV free wall and septum using three-dimensional speckle tracking echocardiography. (**A**) Setting reference points; (**B**,**C**) RV endocardial border tracking; (**D**) Longitudinal strain of RV free wall and septum were automatically generated.

**Figure 3 diagnostics-12-00445-f003:**
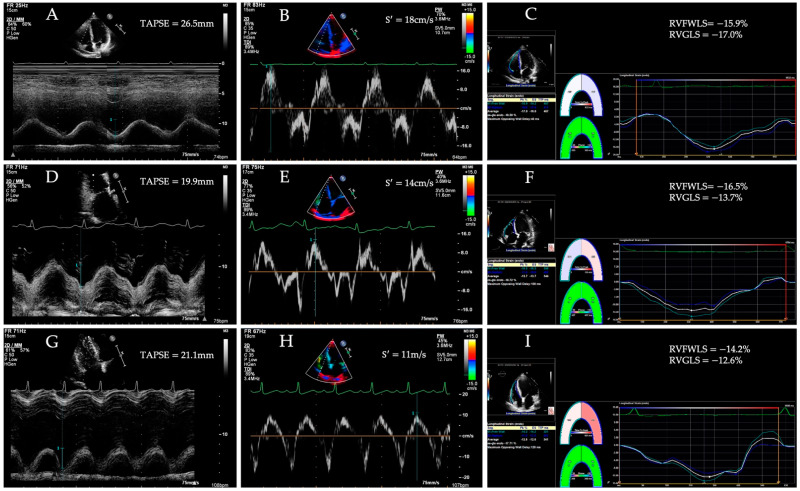
Tricuspid annular plane systolic excursion (TAPSE), systolic velocity of the tricuspid annulus (S’), longitudinal strain of the RV free wall and RV global longitudinal strain in patients with HFpEF (**A**–**C**), HFmrEF (**D**–**F**) and HFrEF (**G**–**I**).

**Table 1 diagnostics-12-00445-t001:** RV longitudinal strain in HFpEF.

Reference	Sample Size	Age (Years), Mean ± SD	Study Design	LVEF (%)	RVGLS (%)	RVFWLS (%)	Software	Device
Liu et al. [78]	86	71.6±11.3	Retrospective		2D-STE-RVLSbas: −25.51±3.92 2D-STE-RVLmid: −21.12±3.91 2D-STE-RVLSapi: −16.92±3.68		Qlab 8.1	Philips Epiq 7C
Morris et al. [79]	201	71.2±10.1	Retrospective	59 ± 7.2	2D-STE-RVGLS: −14.41±3.80		EchoPac 6.1	Vivid-7 (GE Healthcare, Horten, Norway)
Lejeune et al. [80]	149	78±9	Prospective		2D-STE-RVGLS: −21.7±4.9	2D-STE-RVFWLS: −24.2±6.3	TomTec Imaging Systems, Unterschleissheim, Germany	Vivid-7 (GE Healthcare, Horten, Norway)
Meng et al. [81]	81	61 ± 12 *^a^* 61 ± 13 *^b^*	Prospective	66 ± 5 *^a^* 64 ± 4 *^b^*		2D- STE-RVFWLS: −21 ± 3 *^a^* −20 ± 4 *^b^* 3D- STE-RVFWLS: −25 ± 5 *^a^* −23 ± 4 *^b^*	2D Cardiac Performance Analysis 1.2; TOMTEC Imaging Systems GmbH, Unterschleissheim, Germany	Philips iE33; Philips Medical Systems, Andover, MA, USA
Kucukseymen et al. [82]	203	64 ± 12	Retrospective	61 ± 8 *^a^* 59 ± 6 *^b^*		CMR-FT--RVFWLS: −19.6 ± 4.4 *^a^* −14.6 ± 4.6 *^b^*	cvi42; v5.11, Circle Cardiovascular Imaging, Calgary, AB, Canada	1.5T scanner (Phillips Achieva, Best, The Netherlands)
Kammerlander et al. [83]	206	71 ± 8		68 ± 10 *^c^* 59 ± 10 *^d^*	CMR-FT--RVGLS: −11.8 ± 2.2 *^c^* −4.6 ± 3.2 *^d^*		cmr42; Circle Cardiovascular Imaging, Calgary, AB, Canada	1.5-T system (Avanto FIT; Siemens Medi-cal Solutions, Erlangen, Germany)

*^a^* Patients without event (cardiovascular death, hospitalization for acute HF, heart transplantation, intra-aortic balloon pump implantation, and ventricular assist device implantation); *^b^* Patients with event (cardiovascular death, hospitalization for acute HF, heart transplantation, intra-aortic balloon pump implantation, and ventricular assist device implantation); *^c^* RVGLS: −8.5% or less; *^d^* RVGLS: greater than −8.5%.

**Table 2 diagnostics-12-00445-t002:** RV Longitudinal Strain in HFrEF.

Reference	Sample Size	Age (Years), Mean ± SD	Study Design	LVEF (%)	RVGLS (%)	RVFWLS (%)	Software	Device
Houard et al. [13]	266	60 ± 14	Retrospective	23 ± 7	2D-STE-RVGLS: −18.0 ± 4.9 CMR-FT-RVGLS: −11.8 ± 4.3	2D-STE-RVFWLS: −18.7 ± 6.6	4.6 version; TOMTEC Imaging Systems, Unters chleißheim, Germany Segment version 2.2 (Medviso, Lund, Sweden)	Sonus 7500 or iE33 ultrasound systems (Philips Medical Systems, And over, Massachusetts 1.5-T or 3.0-T systems (Intera CV and Achieva, Philips Medical Systems, Best, The Netherlands
Iacoviello et al. [14]	332	64 ± 14	Prospective	33 ± 9	2D-STE-RVGLS: −14.6 ± 4.6	2D-STE-RVFWLS: −21.5 ± 6.2	Echo- PAC PC version; GE Vingmed Ultra-sound	Vivid 7 (GE Vingmed Ultrasound, General Electric, Milwaukee, WI, USA)
Motoki et al. [84]	171	57 ± 14	Retrospective	25 ± 6	2D-STE-RVGLS: −12.4 ± 5.5 *^a^* −10.5 ± 5.1 *^b^*		Velocity Vector Imaging [VVI]; Siemens Medical Solutions USA, Inc.	Acuson Sequoia (Siemens Medical Solutions USA, Inc., Malvern, PA, USA)
Carluccio et al. [85]	288	66 ± 11	Prospective	Median [interquartile range]): 30(25−35)	2D-STE-RVGLS (median [interquartile range]): −15.1[−17.9; −10.8] *^a^* −11.3[−15.1; −8.2] *^b^*	2D-STE-RVFWLS (median [interquartile range]): −20.2[−24.3; −15.6] *^a^* −15.0[−20.0; −11.0] *^b^*	EchoPac 113, General Electric-Vingmed	Vivid 7, Vivid S6, General Electric-Vingmed, Horton, Norway
Cameli et al. [86]	47	57.4 ± 8.1 *^c^* 57.8 ± 7.8 *^d^*	Cross-sectional	25.2 ± 4.5 *^c^* 24.9 ± 4.7 *^d^*	2D-STE-RVGLS: −18.1 ± 2.1 *^c^* −10.9 ± 1.8 *^d^*	2D-STE-RVFWLS: −20.9 ± 2.8 *^c^* −10.6 ± 1.9 *^d^*	EchoPac, GE, Milwaukee, WI, USA	
Houard et al. [87]	20	63 ± 17	Prospective	33 ± 8	2D-STE-RVGLS: −19 ± 4 CMR-FT-RVGLS: −11 ± 6	2D-STE-RVFWLS: −21 ± 6	Tomtec Software (4.6. Version; Tomtec Imaging Systems, Germany) CVI-42 software (Circle CV, Montreal, QC, Canada).	Philips EPIQ 7 ultrasound system (Philips Medical. Systems, Andover, MA, USA) 3T scanner (Achieva, Philips Medical Systems, Best, The Netherlands).

*^a^* Patients without event (cardiovascular death, hospitalization for acute HF, heart transplantation, intra-aortic balloon pump implantation, and ven-tricular assist device implantation); *^b^*Patients with event (cardiovascular death, hospitalization for acute HF, heart transplantation, intra-aortic balloon pump implantation, and ventric-ular assist device implantation); *^c^* RV stroke work index ≥ 0.25 mmHg/Lm^2^; *^d^* RV stroke work index < 0.25 mmHg/Lm^2^.

**Table 3 diagnostics-12-00445-t003:** RV longitudinal strain in HF patients with preserved traditional RV function parameters.

Reference	Sample Size	Age (Years), Mean ± SD	Study Design	LVEF (%)	RVGLS (%)	RVFWLS (%)	Software	Device
Morris et al. [89]	218 *^a^* 208 *^b^*	72.0 ± 10.5 *^a^* 67.4 ± 14.1 *^b^*	Prospective	61.9 ± 6.1 *^a^* 35.4 ± 9.6 *^b^*	2D-STE-RVGLS: −20.7 ± 4.0 *^a^* −15.3 ± 4.7 *^b^*	2D-STE-RVFWLS: −24.6 ± 5.1 *^a^* −19.0 ± 5.8 *^b^*	Echo-Pac 113, GE	Vivid 7 or E9 (GE Healthcare)
Carluccio et al. [24]	200	66 ± 11	Prospective	30 ± 7		2D-STE-RVFWLS: −20.9 ± 5.9 *^c^* −16.7 ± 5.6 *^d^*	EchoPac 112.1.5; General Electric-Vingmed	(Vivid 7, Vivid S6; General Electric Vingmed, Horton, Norway

*^a^* Patients with HFpEF; *^b^* Patients with HFrEF; *^c^* Patients without event (cardiovascular death, hospitalization for acute HF, heart transplantation, intra-aortic balloon pump implantation, and ventricular assist device implantation); *^d^* Patients with event (cardiovascular death, hospitalization for acute HF, heart transplantation, intra-aortic balloon pump implantation, and ventricular assist device implantation).

**Table 4 diagnostics-12-00445-t004:** RV longitudinal strain in acute HF.

Reference	Sample Size	Age (Years), Mean ± SD	Study Design	LVEF (%)	RVGLS (%)	RVFWLS (%)	Software	Device
Hamada-Harimura et al. [90]	618	72 ± 13	Prospective	46 ± 16	2D-STE-RVGLS: −11.9 ± 5.1 *^a^* −11.0 ± 5.1 *^b^*	2D-STE-RVFWLS: −15.5 ± 5.8 *^a^* −13.5 ± 5.9 *^b^*	TomTec Imaging System, Munich, Ger-many	GE Healthcare (Milwaukee, WI), Philips (Andover), or Toshiba Medical Systems (Tochigi, Japan).
Yao et al. [91]	38	61 ± 10 *^c^* 58 ± 10 *^d^*	Prospective	44.5 ± 9.7 *^c^* 46.7 ± 11.5 *^d^*	2D-STE-RVGLS: −10.5 ± 4.2 *^c^* −14.1 ± 5.1 *^d^*	RVFWLSapi: −12.0 ± 8.5 *^c^* −20.1 ± 5.9 *^d^* 2D-STE-RVFWLSmid: −10.8 ± 5.1 *^c^* −17.0 ± 5.6 *^d^* 2D-STE-RVFWLSbas: −9.7 ± 7.3 *^c^* −13.5 ± 5.6 *^d^*	GE EchoPac, version 113	GE Vivid q
Borovac et al. [92]	42	Median [inter-quartile range]): 71.5(62−76)	Cross-sectional	39.1 ± 16.0	2D-STE-RVFWLS: −11.6 ± 2.8 *^e^* −21.3 ± 3.7 *^f^*		EchoPac PC, version 112; GE Medical Systems, Milwaukee, WI, USA	VividTM 9 ultrasound system (GE Medical Systems, Milwaukee, WI, USA)
Park et al. [93]	1824	70.4 ± 13.8	Retrospective	39.3 ±15.2	2D-STE-RVGLS: Group 1 (LVGLS ≥ 9% and RVGLS ≥ 12%): −17.6 ± 4.7 Group 2 (LVGLS ≥ 9% and RVGLS < 12%): −8.2 ± 2.6 Group 3 (LVGLS < 9% and RVGLS ≥ 12%): −15.8 ± 3.6 Group 4 (LVGLS < 9% and RVGLS < 12%) −6.6 ± 2.9		TomTec (Image Arena 4.6)	commercial echocardio-graphic machines

*^a^* Patients without event (cardiovascular death, hospitalization for acute HF, heart transplantation, intra-aortic balloon pump implantation, and ventricular assist device implantation); *^b^* Patients with event (cardiovascular death, hospitalization for acute HF, heart transplantation, intra-aortic balloon pump implantation, and ventricular assist device implantation); *^c^* Patients with right ventricular myocardial infarction.; *^d^* Patients with left ventricular inferior myocardial infarction; *^e^* Patients in the subgroup below RVFWS median (RVFWS median: −16.5%, interquartile range: (−20.1)–(−11.2)%); *^f^* Patients in the subgroup above RVFWS median (RVFWS median: −16.5%, interquartile range: (−20.1)–(−11.2)%).

**Table 5 diagnostics-12-00445-t005:** RV longitudinal strain in non-ischemic dilated cardiomyopathy.

Reference	Sample Size	Age (Years), Mean ± SD	Study Design	LVEF (%)	RVGLS (%)	RVFWLS (%)	Software	Device
Vîjîiac et al. [97]	50	61 ± 14	Prospective	25 ± 7	2D-STE-RVGLS: −14.30 ± 5.20 *^a^* −10.50 ± 4.50 *^b^*	2D-STE-RVFWLS: −17.50 ± 7.10 *^a^* −12.90 ± 8.70 *^b^*	EchoPAC—Q Analysis package	Vivid E9 (GE Vingmed, Horten, Norway)
Ishiwata et al. [98]	109	44 ± 14	Retrospective	23.8 ± 7.3 *^a^* 18.9 ± 7.5 *^b^*	2D-STE-RVGLS (median [inter-quartile range]): −13.8[−21.1; −10.8] *^a^* −9.9[−14.5; −7.0] *^b^*		2D Strain Analy-sis; TOMTEC Imaging System, Unterschleissheim, Germany	
Liu et al. [99]	192	53 ± 14	Prospective	22.37 ± 9.75	CMR-FT-RVpGLS: −10.49 ± 5.16		CVI42 software (Version 5.6.3 Circle Cardiovascu-lar Imaging, Calgary, AB, Canada)	3.0T scanner (Magnetom Verio; Siemens AG Healthcare, Erlangen, Germany or MR750W, General Electric Healthcare, Waukesha, WI, USA)

*^a^* Patients without event (cardiovascular death, hospitalization for acute HF, heart transplantation, intra-aortic balloon pump implantation, and ventricular assist device implantation); *^b^* Patients with event (cardiovascular death, hospitalization for acute HF, heart transplantation, intra-aortic balloon pump implantation, and ventricular assist device implantation).

**Table 6 diagnostics-12-00445-t006:** RV longitudinal strain and myocardial fibrosis in patients with end-stage HF.

Reference	Sample Size	Age (Years), Mean ± SD	Study Design	LVEF (%)	RVGLS (%)	RVFWLS (%)	Software	Device
Cordero-Reyes et al. [100]	20	53 ± 13	Cross-sectional	30.0 ± 3.5		2D-STE-RVFWLS (median [inter-quartile range]): −5.6[−7.6; −2.8]		syngo Velocity Vector Imaging, Siemens Healthcare, Malvern, Pennsylvania
Lisi et al. [101]	27	53.7 ± 4.6	Cross-sectional	22.3 ± 2.4		2D-STE-RVFWLS: −15.3 ± 4.7	EchoPac, GE, Waukesha, WI, USA	Vivid 7, GE Medical System echo-cardiograph (Horten, Norway)
Tian et al. [102]	102	44.41 ± 13.51 *^a^* 43.91 ± 17.49 *^b^* 44.65 ± 17.02 *^c^*	Cross-sectional	26.24 ± 6.92 *^a^* 23.82 ± 6.14 *^b^* 23.92 ± 4.72 *^c^*		2D-STE-RVFWLS: −14.32 ± 3.57 *^a^* −12.84 ± 3.69 *^b^* −9.78 ± 3.03 *^c^* 3D-STE-RVFWLS: −13.93 ± 2.67 *^a^* −12.33 ± 3.17 *^b^* −8.27 ± 3.10 *^c^*	TomTec	Philips Epiq 7C

*^a^* Patients with mild myocardial fibrosis; *^b^* Patients with moderate myocardial fibrosis; *^c^* Patients with severe myocardial fibrosis.

**Table 7 diagnostics-12-00445-t007:** RV longitudinal strain in RV failure following LV assist device implantation.

Reference	Sample Size	Age (Years), Mean ± SD	Study Design	LVEF (%)	RVGLS (%)	RVFWLS (%)	Software	Device
Dufendach et al. [15]	137	median [inter-quartilerange]: 59.5[52; 56] ^a^ 55[45; 63] ^b^	Retrospective		2D-STE-RVGLS (median [inter-quartilerange]): −7.315[−10.45; −4.51] *^a^* −6.200[−8.44; −3.70] *^b^*	2D-STE-RVFWLS (median [inter-quartile range]): −7.835[−11.27; −4.77] *^a^* −5.490[−8.60; −4.17] *^b^*	TomTec	
Grant et al. [107]	117	median [inter-quartile range]: 58[47.5; 65]	Retrospective	median [inter-quartile range]: 15[10; 20] ^a^ 15[10; 25] *^b^*		2D-STE-RVFWLS: (median [inter-quartile range]): −12.2[−14.9; −9.5] *^a^* −9.0[−11.4; −7.3] *^b^*	Velocity Vector Imag-ing, Siemens AG, Erlangen, Germany	
Cameli et al. [108]	10	66.4 ± 5.1 *^a^* 65.8 ± 4.8 *^b^*	Retrospective	25.2 ± 4.5 *^a^* 24.9 ± 4.7 *^b^*	2D-STE-RVGLS: −14.1 ± 2.1 *^a^* −8.9 ± 1.8 *^b^*	2D-STE-RVFWLS: −15.5 ± 3.6 *^a^* −9.2 ± 1.9 *^b^*	EchoPac, General Electric Healthcare	Vivid 7, GE Vingmed, Horten, Norway
Magunia et al. [104]	26	64 ± 13 *^a^* 58 ± 30 *^b^*	Retrospective	≤20 (95.2%) *^a^*, 21−30(4.8%) *^a^* ≤20(100%) *^b^*		3D-STE-RVFWLS: −13.2 ± 4.7 *^a^* median [inter-quartile range]): −6[−8.8; −4.3] *^b^*	Tomtec Image Arena and Tomtec 2D Cardiac Performance Analysis, Tomtec Imaging Systems GmbH, Unterschleissheim, Germany	Philips iE33-system, X7-2t Matrix probe, Philips Healthcare Inc., Andover, MA, USA

*^a^* Patients without RV failure; *^b^* Patients with RV failure.

**Table 8 diagnostics-12-00445-t008:** RV longitudinal strain in congenital heart diseases.

Reference	Sample Size	Age (Years), Mean ± SD	Study Design	LVEF (%)	RVGLS (%)	RVFWLS (%)	Software	Device
Almeida-Morais et al. [115]	42	32 ± 8	Prospective	58 ± 8	2D-STE-RVGLS: −16.2 ± 3.7		EchoPAC Program; GE Healthcare	Vivid-E9 (GE HealthcareTechnology, General Electric Vingmed Ultrasound, Horten, Norway)
Timóteo et al. [116]	26	30 ± 9	Retrospective		2D-STE-RVGLS (median [inter-quartile range]): −13.0[−15.2; −9.4] *^a^* −20.9[−23.6; −18.9] *^b^*	2D-STE-RVFWLS (median [inter-quartile range]): −13.2[−14.3; −10.5] *^a^* −21.8[−25.3; −17.9] *^b^*	EchoPACTM, GE Healthcare	Vivid 7TM and Vivid 9TM, GE Healthcare
Steinmetz et al. [117]	30	mean: 26.3	Prospective		CMR-FT-RVGLS: −13.48 ± 6.26		TomTec Imaging Systems, 2D CPA MR, Cardiac Performance Analysis, Version 1.1.2.36, Unterschleissheim, Germany	1.5 Tesla MRT-“Symphony“ -scanner (Siemens Medical Systems, Erlangen, Germany

*^a^* Systemic RV; *^b^* Pulmonary RV.

**Table 9 diagnostics-12-00445-t009:** RV longitudinal strain in chemotherapy cardiotoxicity.

Reference	Sample Size	Age (Years), Mean ± SD	Study Design	LVEF (%)	RVGLS (%)	RVFWLS (%)	Software	Device
Wang et al. [119]	61	50.8 ± 12.1	Prospective	61.4 ± 4.8 *^a^* 59.3 ± 4.3 *^b^* 57.5 ± 5.7 *^c^* 55.2 ± 4.2*^d^*	2D-STE-RVGLS: −22.5 ± 3.6 *^a^* −20.8 ± 3.2 *^b^* −20.2 ± 3.3 *^c^* −19.8 ± 3.5 *^d^*	2D-STE-RVFWLS: −25.8 ± 3.8 *^a^* −25.1 ± 3.5 *^b^* −23.8 ± 3.6 *^c^* −23.2 ± 3.4 *^d^*	QLAB version8.1; Philips Medical System	iE33 scanner from Philips (Bothell, WA, USA)
Keramida et al. [120]	101	54.3 ± 11.4	Retrospective	61.8 ± 4.1 *^a^* 59.6 ± 5.0 *^e^* 59.2 ± 6.2 *^f^* 58.4±5.7 *^g^* 60.0 ± 6.7 *^h^*	2D-STE-RVGLS: −21.3 ± 4.5 *^a^* −20.7 ± 4.2 *^e^* −19.6 ± 5.2 *^f^* −20.1 ± 4.0 *^g^* −20.1 ± 3.9 *^h^*	2D-STE-RVFWLS: −21.4 ± 4.4 *^a^* −20.9 ± 4.6 *^e^* −19.7 ± 5.6 *^f^* −20.6 ± 4.3 *^g^* −20.5 ± 4.5 *^h^*	TomTec Imaging Systems, Unterschleissheim, Germany	GE Vivid E9; and Philips iE33

*^a^* Echocardiographic parameters in baseline; *^b^* Echocardiographic parameters after the third cycle; *^c^* Echocardiographic parameters in baseline, after the sixth–-eighth cycle; *^d^* Echocardiographic parameters during follow-up; *^e^* Echocardiographic parameters at the third 3rd month of follow-up; *^f^* Echocardiographic parameters at the sixth 6th month of follow-up; *^g^* Echocardiographic parameters at the ninth 9th month of follow-up; *^h^* Echocardiographic parameters at the twelfth 12th month of follow-up.

## Data Availability

Not applicable.

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
