# Peer review of "Right Ventricular Longitudinal Strain in Patients with Heart Failure"

_diagnostics, 2022, doi:10.3390/diagnostics12020445_

Round 1
Reviewer 1 Report
The present study reviewed clinical significance of right ventricular longitudinal strain (RVLS) in heart failure. Previous review have already demonstrated usefulness of RVLS in heart failure and pulmonary hypertension. This review included heart failure with preserved left ventricular ejection fraction and acute heart failure. However, there are no important data of those each references in the present review. Please summarize the subjects, the study design, results, and limitations of main references of each category in tables according to the approach of systematic review. These approaches are important from the viewpoints of critical appraisal to conclude the usefulness of RVLS in heart failure.
Author Response
Point 1: The present study reviewed clinical significance of right ventricular longitudinal strain (RVLS) in heart failure. Previous review have already demonstrated usefulness of RVLS in heart failure and pulmonary hypertension. This review included heart failure with preserved left ventricular ejection fraction and acute heart failure. However, there are no important data of those each references in the present review. Please summarize the subjects, the study design, results, and limitations of main references of each category in tables according to the approach of systematic review. These approaches are important from the viewpoints of critical appraisal to conclude the usefulness of RVLS in heart failure.
Response 1: We are grateful for your comment. Based on your suggestion, we have added tables that summarized the sample size, age, study design, results, software and device of the main references for each category in a systematic review approach.
Reviewer 2 Report
The manuscript is well written. The authors summarize clearly the different echocardiographic methods used to evaluate the right ventricular function (from tissue Doppler Imaging to the right ventricle speckle tracking echocardiography). They summarize the prognostic role of the right ventricular longitudinal strain assessed by 2D and 3D speckle-tracking echocardiography in patients with heart failure both with preserved ejection fraction and with reduced ejection fraction. However, I think that this review does not add any novelties in the field of multi-modality imaging of the right ventricle in patients with heart failure. Few studies have been analyzed on the 3D right ventricular longitudinal strain. Furthermore, the authors do not examine the prognostic role of the right ventricular longitudinal strain analyzed by the cardiac magnetic resonance feature tracking. The authors only analyze the 2D and 3D right ventricular longitudinal strain in patients with heart failure and they do not refer to the evaluation of the right ventricular contractile reserve (the new echocardiographic parameter is very important in patients with heart failure). In the context of heart failure, the authors make no differentiation about the prognostic role of the right ventricular longitudinal strain in various non-ischemic dilated cardiomyopathy. In addition, the authors analyze the role of the right ventricular longitudinal strain in LVAD patients by providing practical information; they could provide practical information on the application and prognosis of the right ventricular longitudinal strain in other diseases (congenital heart disease and chemotherapy cardiotoxicity).
Author Response
Point 1: The manuscript is well written. The authors summarize clearly the different echocardiographic methods used to evaluate the right ventricular function (from tissue Doppler Imaging to the right ventricle speckle tracking echocardiography). They summarize the prognostic role of the right ventricular longitudinal strain assessed by 2D and 3D speckle-tracking echocardiography in patients with heart failure both with preserved ejection fraction and with reduced ejection fraction. However, I think that this review does not add any novelties in the field of multi-modality imaging of the right ventricle in patients with heart failure. Few studies have been analyzed on the 3D right ventricular longitudinal strain. Furthermore, the authors do not examine the prognostic role of the right ventricular longitudinal strain analyzed by the cardiac magnetic resonance feature tracking.
Response 1: Thank you very much for your comments. We agree that right ventricular longitudinal strain assessed by cardiac magnetic resonance feature tracking provides significantly value for predicting the outcome in patients with cardiovascular diseases. And we have added the prognostic value of right ventricular longitudinal strain derived from cardiac magnetic resonance feature tracking in HFpEF, HFrEF, non-ischemic dilated cardiomyopathy and congenital heart diseases.
Point 2: The authors only analyze the 2D and 3D right ventricular longitudinal strain in patients with heart failure and they do not refer to the evaluation of the right ventricular contractile reserve (the new echocardiographic parameter is very important in patients with heart failure).
Response 2: We thank the reviewer for this valuable comment. According to your suggestion, we had added the value of right ventricular contractile reserve in patients with heart failure, and the relevant references have been added in this revised manuscript.
Point 3: In the context of heart failure, the authors make no differentiation about the prognostic role of the right ventricular longitudinal strain in various non-ischemic dilated cardiomyopathy.
Response 3: Thank you very much for your comments. We have added the utility of right ventricular longitudinal strain in non-ischemic dilated cardiomyopathy. The relevant references can be seen in the revised manuscript.
Point4: In addition, the authors analyze the role of the right ventricular longitudinal strain in LVAD patients by providing practical information; they could provide practical information on the application and prognosis of the right ventricular longitudinal strain in other diseases (congenital heart disease and chemotherapy cardiotoxicity).
Response 4: Thank you very much for your comments. We also agree with your comment that right ventricular longitudinal strain can provide significant prognostic value in other cardiovascular diseases, and we had added the usefulness of right ventricular longitudinal strain in patients with congenital heart diseases and in the chemotherapy cardiotoxicity.
Reviewer 3 Report
The review is clearly written and comprehensive.
Nothing to correct
Author Response
Point 1: The review is clearly written and comprehensive. Nothing to correct
Response 1: Thank you for your nice comment.
Round 2
Reviewer 1 Report
I have no further comments.
Author Response
Thank you.
Reviewer 2 Report
The authors answered the questions very well. I think that the manuscript can be accepted.
Author Response
Thank you.